# LATTICE REPRESENTATION LEARNING

## ABSTRACT

We introduce the notion of *lattice representation learning*, in which the representation for some object of interest (e.g. a sentence or an image) is a lattice point in an Euclidean space. Our main contribution is a result for replacing an objective function which employs lattice quantization with an objective function in which quantization is absent, thus allowing optimization techniques based on gradient descent to apply; we call the resulting algorithms *dithered stochastic gradient descent* algorithms as they are designed explicitly to allow for an optimization procedure where only local information is employed. We also argue that a technique commonly used in Variational Auto-Encoders (Gaussian priors and Gaussian approximate posteriors) is tightly connected with the idea of lattice representations, as the quantization error in good high dimensional lattices can be modeled as a Gaussian distribution. We provide experimental evidence of the potential of using lattice representations by implementing a VAE that employs a rectangular lattice and is trained using our results, and contrasting its performance with that of a Gaussian VAE.

## 1 PRELIMINARIES

With a few notable exceptions, the majority of the practical research in representation learning assumes that the representation of the objects of interest (sentences, images, audio signals, etc.) are vectors of real numbers, as this allows us to use powerful optimization algorithms such as variants of gradient descent in the training of computational networks which encode objects into such representations and then use those representations in downstream tasks. Yet, the idea of representing objects using discrete structures (for example, through categorical variables or through the use of quantization of otherwise real valued representations) is rather enticing: sometimes we might inherently believe discrete representations to be the right way to model objects, or may want to use such representations in settings such as reinforcement learning and planning, where discrete actions are important. A classical result by Lindsay (1983) for maximum likelihood learning of mixture models tells us that the optimal mixing distribution can be chosen to be discrete (and in fact, the discrete set need not be larger than the amount of training data); this result implies also that the optimal associated "approximate posterior" (when seen as a variational inference problem) in fact can be chosen so that it produces discrete representations. The main difficulty associated with discrete representations is that it is not straightforward to train networks that produce and use them because either there is no meaningful sense in which differentiation can be used directly (e.g. in the case of categorical variables), or in the case of quantization, the associated gradient is zero almost everywhere. In spite of these difficulties, notable progress has been made. For example, for categorical variables, Jang et al. (2017), and Maddison et al. (2017) proposed essentially the same idea, under the names of Gumbel-Softmax and the Concrete Distribution, respectively. This idea, further improved by Tucker et al. (2017), uses a continuous approximation to a "one-hot" encoded categorical distribution which can be learned by appropriately setting a parameter controlling the fidelity of the approximation. For the setting where a continuous representation is quantized to obtain the discrete one, an important development is the idea of *straight-through* estimation (Bengio et al., 2013), in which quantization is applied in the "forward" direction, but replaces quantization with the identity operator when performing the "backwards" differentiation step - see also recent work attempting to provide theoretical justification of straight-through estimation (Yin et al., 2017). Of particular relevance to our work is VQ-VAE (van den Oord et al., 2017) in which general vector quantization is used, together with straight-through estimation, to train the network and the vector quantizers. In fact, the entirety of this paper can be seen as a type of VQ-VAE where the vector

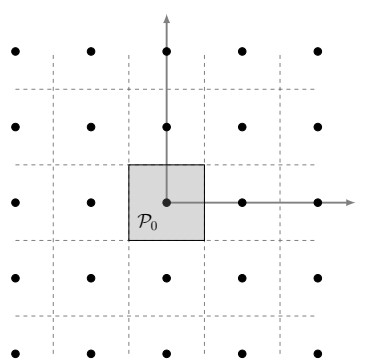 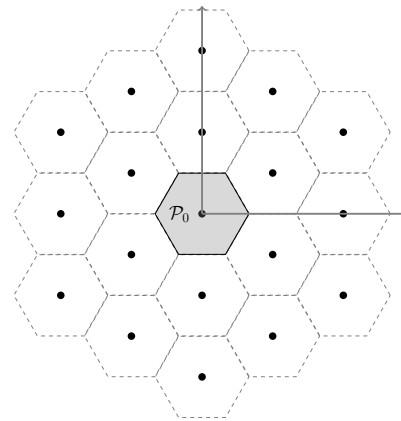

Figure 1: Two examples of two dimensional lattices. In the left, a cubic lattice with basis vectors $\mathbf{b}_1 = (0, \Delta)$, $\mathbf{b}_2 = (\Delta, 0)$. In the right, a hexagonal lattice with basis vectors $\mathbf{b}_1 = (0, 2)$, $\mathbf{b}_2 = (\sqrt{3}, 1)$. Also shown for each lattice is the cell $\mathcal{P}_0$, containing the origin. The parameter of the cubic lattice $\Delta$ has been set so that the areas of both lattice cells are identical.

quantizers are highly structured instead of being free parameters in a network; as we will see, the advantages of introducing structure are that we can bring to bear analytical tools that give insight not only on how to train these networks with an equivalent continuous version, but also shed light on how we can approximate the performance of continuous Gaussian VAEs in a systematic way.

In this article, we study the possibility of using lattices for object representations by borrowing from theoretical developments in information and coding theory (Zamir et al., 2014), in particular as they apply to lossy source coding problems (Shannon, 1959) - see Figure 1 for an example of two lattices. In our article, a lattice is defined as the set of all integral linear mixtures of a given set of basis vectors for an Euclidean space. Lattices have been long studied in information theory as they provide powerful means for building structured codes which are analytically tractable and have good space packing and covering properties, which are useful in channel coding and lossy source coding applications, respectively (Zamir et al., 2014). We note that it has long been known that machine learning and lossy compression are related to each other; for example, we refer the reader to the classical work on the *information bottleneck* (Tishby et al., 1999; Tishby & Zaslavsky, 2015; Shwartz-Ziv & Tishby, 2017; Slonim & Weiss, 2002), the connection between maximum likelihood estimation and rate distortion explored by Banerjee et al. (2004), Lastras-Montaño (2019), and implicitly, Rose (1998), Neal & Hinton (1998) and a line of research on autoencoders (Giraldo & Príncipe, 2013; Alemi et al., 2018; Higgins et al., 2017). Our work adds to this general line of research by adopting more specific structures - in this case, lattices - from the information theory field and applying them to machine learning problems.

Our primary contribution will be to provide the necessary theory to train computational networks that employ lattices and *dithered quantization*, leading to a class of algorithms which we call *dithered stochastic gradient descent* algorithms. Dithering refers to the act of adding random noise to a quantized signal (e.g. an image or a time series) for the purposes of diminishing the effect that quantization has on it. The main reason dithering is important to our problem can be traced to a fundamental result in the study of lattices in information theory called the "Crypto-Lemma" (see Zamir & Feder (1996), G. D. Forney (2004)) which to our knowledge, had not been used before to optimize, with gradient techniques, computational networks that employ quantization in a provably correct way. We will additionally connect the task of lattice representation learning with a well known technique in generative modeling using variational autoencoders; specifically, Gaussian approximate posteriors (Kingma & Welling, 2014). In our experimental section, we will demonstrate how we can use our results to train a continuous VAE that can be re-interpreted, through our main result, as a discrete VAE that uses a finite dimensional lattice.

## 2 PROBLEM SETUP

To make our work concrete, we will illustrate it in the context of a standard encoder/decoder architecture; for a given object $d \in \mathcal{D}$ (where $\mathcal{D}$ is the space of all possible objects), it's representation is produced using an encoder:

$$e(d)$$

which is then decoded using a decoder:

$$g(e(d)).$$

We pause to remark that there is a subtle difference between the way in which the information theory and machine learning communities use the words "encoder" and "decoder"; the reader needs to be aware of this difference so as to not get lost in this article. In information theory, an encoder/decoder is almost always a means to represent and retrieve information *digitally*, in particular, there is a significant emphasis on the efficiency of such representations. In machine learning, an encoder's output is generally thought of as a continuous vector (of course, as we stated earlier, there are notable exceptions to this); the fact that such a continuous vector is represented digitally (in the form of a computer representation of a floating point number) is in a sense an afterthought. Significantly more important in machine learning is the capability of learning parameters for an encoder and decoder through training examples, in which case the assumption of continuity, and more generally, differentiability is much more important.

Our work lies in the middle of the two fields. We are interested in a machine learning application, but want to emphasize the representation cost for the objects being encoded as a first class metric to be optimized for. In our work, the representations are discrete, this is for any $d \in \mathcal{D}$, $e(d)$ belongs to a discrete space, which we denote as $\mathcal{X}$. To each element $x \in \mathcal{X}$ we assign a probability $p(x)$, so that we can talk about a *code* for the representation space $\mathcal{X}$. In this setting, a code is a mechanism for mapping $x \in \mathcal{X}$ to $l(x)$ bits (where $l(x)$ stands for the *code bit length*) so that $x$ can be recovered, without loss, from the code for $x$ of length $l(x)$. It is well known (Cover & Thomas, 2006) that for any given code one can construct a probability distribution using the equation $p(x) = 2^{-l(x)}$. It is also known that for a given distribution $\{p(x)\}$, the ideal choice for $l(x)$ (in units of bits) is $-\log_2 p(x)$, in the sense of minimizing the number of expected bits under the assumption that the $x$ are being generated with a distribution $\{p(x)\}$. It is also known that such ideal code length in general can only be achieved with *block codes* which compress multiple elements of $\mathcal{X}$ simultaneously. In our work, we will be indifferent to this - we will simply use $l(x) = -\log p(x)$ for the code length; in fact for our theoretical development we will be using the unit of *nats* (natural logarithms) as it is more convenient.

Assume that $D$ is a random quantity uniformly distributed over the training data $\{d_1, \cdots, d_n\}$. Then the average representation cost for the training data is

$$-E_D \log P(e(D)) \tag{1}$$

This is one of the metrics in our work. The other metric is free to be specified by the designer of the architecture. Typically, this will be of the form

$$E_D \left[ \ell(D, g(e(D))) \right] \tag{2}$$

where $\ell$ is some loss function. This covers both unsupervised and supervised settings; in the latter, we assume that the loss function also includes the label for $D$. It also includes variational autoencoders when they are trained using the Evidence Lower Bound (ELBO).

One way to create discrete representations $e(D)$ is to quantize an otherwise continuous representation $e_c$:

$$e(D) = K(e_c(D))$$

where $K$ denotes a quantization operation; for example, $K$ could use the uniform quantizer with spacing $\Delta$ applied to every one of the $m$ dimensions. The encoder $e()$, the decoder $g()$ and the representation code $-\log p()$ all participate in the end-to-end objective function

$$E_D \left[ \ell(D, g(K(e_c(D)))) \right] + \lambda E_D \left[ \log \frac{1}{P(K(e_c(D)))} \right] \tag{3}$$

where $\lambda > 0$ is a parameter that controls the importance of the representation cost for the overall optimization process.

In a nutshell, the problem to be solved is to find a theoretically sound basis for the design of good quantizers $K$ that can be trained in an end-to-end computational network. As stated earlier, this can be seen as an attempt to bring additional mathematical rigor to the VQ-VAE concept.

## 3 DITHERED STOCHASTIC GRADIENT DESCENT

### 3.1 MATHEMATICAL PRELIMINARIES

The principles behind the use of lattices in information theory can be found in the excellent book of Zamir et al. (2014) . We borrow from this exposition in here in order to make this work self contained. Let $b_1, \cdots, b_m$ be a basis for $\mathbb{R}^m$, this is, $m$ linearly independent column vectors in $\mathbb{R}^m$, and define $B$ to be the matrix obtained using these basis vectors as its columns: $B = [b_1|b_2|\cdots|b_m]$. The lattice $\Lambda(B)$ is defined as the set of all integral linear combinations of the $\{b_1, \cdots, b_m\}$:

$$\Lambda(B) = \{B \cdot \mathbf{i} : \mathbf{i} \in Z^m\}.$$

where $Z^m = \{\cdots, -2, -1, 0, 1, 2, \cdots\}^m$. When clear from the context, we will use $\Lambda$ to refer to a lattice. We define $\mathcal{P}_0(\Lambda)$ to be the set of all points in $\mathbb{R}^m$ whose closest lattice point is the origin.

Given $x \in \mathbb{R}^m$, we define $K_\Lambda(x)$ to be the operator (the "quantizer") that takes $x$ and maps it to the closest element of the lattice $\Lambda$:

$$K_\Lambda(x) = \underset{p \in \Lambda}{\arg\min} \|x - p\|_2 \tag{4}$$

One of the main reasons we are interested in lattices is because of a mathematical tool that can be used in their analysis, which in information theory is colloquially referred to as the "Crypto-Lemma" (Zamir & Feder, 1996), (G. D. Forney, 2004):

**Lemma 1** (Crypto-Lemma). *For a given $m$ dimensional lattice $\Lambda$, let $U$ be uniformly distributed over $\mathcal{P}_0(\Lambda)$. For any $x \in \mathbb{R}^m$, the distribution of the random vector $K_\Lambda(x + U) - U$ is identical to the distribution of $x - U$.*

The Crypto-Lemma will give us the main mechanism for training a computational network using gradient descent algorithms and using such network (during inference) with an explicit quantization step, with a firm theoretical guarantee of equivalence.

A few additional mathematical preliminaries are needed. When taking expectations we will explicitly state the probability law being used in the expectation by placing a random quantity (most of the time, a random vector) as a subindex of the expectation. If two random vectors $A$, $B$ are independent, we will write $A \perp\!\!\!\perp B$. If these random vectors are independent conditional on a third random vector $C$, we will write $(A \perp\!\!\!\perp B | C)$. We will use an upper case $P$ to denote a probability measure, this is, a function that assigns a probability to an event passed as argument. We will use the notation $P_A(a)$ as a summary form of $P([A = a])$; we sometimes will use a random vector as an argument, thus, for example, $-E_A \log P_{\hat{A}}(A)$ can be interpreted as the average cost, in nats, of using a code designed for $\hat{A}$ on the random vector $A$, which is drawn using a (generally) different probability law. We will use the notation $f_A$ to denote density of a continuous random vector $A$. We will use the notation $P_{A|B}$ and $f_{A|B}$ to denote conditional versions of the objects described above. In a slight overload of notation, recall we use $f$ and $e_c$ to denote encoders; the correct usage should be clear from context. If $P_1$ and $P_2$ are two probability measures, we say that $P_1 \ll P_2$ (in words, $P_1$ is absolutely continuous with respect to $P_2$) if for any event $A$ such that $P_2(A) = 0$ then we have $P_1(A) = 0$.

### 3.2 THE MAIN RESULT

Our main contribution will be centered on the following new result:

**Theorem 1** (representation cost for dithered SGD). *Let $\Lambda$ be any lattice. Let $X$ be a random $\mathbb{R}^m$ vector distributed according to $P_X$, and $\hat{X}$ be a random $\mathbb{R}^m$ vector distributed according to $P_{\hat{X}}$, where we assume that $P_X \ll P_{\hat{X}}$. Assume $U$ is uniformly distributed over $\mathcal{P}_0$, and assume that*

$U \perp\!\!\!\perp X$ and $U \perp\!\!\!\perp \hat{X}$. *Define*

$$Z = K_\Lambda(X + U) \qquad (5)$$

$$\hat{Z} = K_\Lambda(\hat{X} + U) \qquad (6)$$

*Then*

$$E_{X,U}\left[\log \frac{1}{P_{\hat{Z}|U}(Z|U)}\right] = E_{X,U}\left[\log \frac{f_U(U)}{f_{\hat{X}-U}(X-U)}\right]. \qquad (7)$$

The proof of this result is based on a dual application of the Crypto-Lemma, and can be found in the Appendix. In the application of result, $P_X$ will be associated with the empirical statistics of the encoding of training data (prior to quantization), and $P_{\hat{X}}$ will be associated with a *prior belief* on the statistics of the encodings of any data, prior to quantization.

Notice that the right hand side of Equation (7) has *no quantization involved*. Notice also that the expression in the expectation does *not* use directly any global statistics about $X$; instead it relies on $f_{\hat{X}-U}$, which is a density that the designer can specify when the statistics of $\hat{X}$ are specified and when the lattice $\Lambda$ is designed. By avoiding the explicit quantization we eliminate the problem resulting from the gradient of a quantized signal being zero almost everywhere, and by ensuring that the expression being optimized does not directly depend on global statistics about $X$, we ensure that stochastic gradient descent is feasible, since it requires that you update parameters based on gradients computed on small batches of training data. For these reasons, we shall call the machine learning algorithms that one derives from Theorem 1 *dithered stochastic gradient descent* algorithms.

Now examine the left hand side of Equation (7), and notice that it *does* involve explicitly quantization: in fact, $Z$ will become the discrete *lattice representation* of our work. Notice the conditioning in $P_{\hat{Z}|U}$ on the dither value $U$. This means that the discrete representation not only depends on the dither $U$, but also is being encoded using a code that depends $U$ as well. Notice also that $-\log P_{\hat{Z}|U}$ may not be the very best code one could be using which is $-\log P_{Z|U}$, in fact

$$E_{X,U}\left[\log \frac{1}{P_{\hat{Z}|U}(Z|U)}\right] = E_{X,U}\left[\log \frac{1}{P_{Z|U}(Z|U)}\right] + D_{KL}\left(Z\|\hat{Z}|U\right)$$

where the former is the "true" best possible representation cost (for a given encoder) and the latter term (a conditional KL divergence which is known to be nonnegative) denotes the excess cost. Thus the representation cost may not be, in a certain sense, "optimum", which is what we pay to make stochastic gradient descent feasible by avoiding having to compute global statistics at every step of the optimization. Having said this, our anecdotal experience is that this is not a significant problem if one allows the parameters of the encoder to also be trained, as then the encoder "adapts" so as to produce representations that approximate the prior that we have set.

A reader knowledgable in the literature of lattices in lossy compression theory may be wondering what is the connection between Theorem 1 and the information theoretic result linking entropy coded dithered quantization and mutual information (Zamir & Feder, 1996). The answer is that Theorem 1 generalizes this result to a setting where we aren't necessarily using the optimum code, which happens to be particularly useful in stochastic gradient descent algorithms.

We now construct the system that we intend to study. We refer the reader to Figure 2, where we illustrate an encoder/decoder architecture coupled to a loss function. In the left hand side, we have a network that employs pre/post dithered quantization of a representation quantized using a lattice $\Lambda$; in the right, there is subtractive dithering but no quantization.

Following Figure 2, define

$$X = e_c(D).$$

With this definition, we can see that in the left hand side of Figure 2, the quantized representation $Z$ matches the definition in Equation (5) of Theorem 1:

$$Z = K_\Lambda(X + U)$$

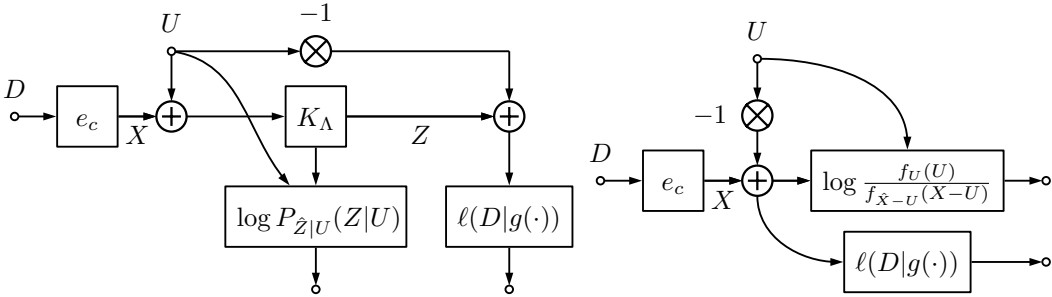

Figure 2: Pre/post dithered quantization in the context of lattice representation learning. The system in the left is used during inference time, the one in the right during training time.

Now in reference to the optimization objective in Equation (3), note that one of the things we need to do is to define a code that will be used to encode $Z$. For this purpose, we will use a code constructed using a distribution $P_{\hat{Z}|U}$. The way we will construct this conditional distribution is also given by the equations in Theorem 1. In particular, the designer will be free to specify any distribution $P_{\hat{X}}$ of their choice, and then construct the random vectors $\hat{X}, U$ so that they are independent random vectors distributed according to $P_{\hat{X}}$ and $P_U$, respectively. Next, as in Equation (6) we define

$$\hat{Z} = K_\Lambda(\hat{X} + U)$$

The distribution $P_{\hat{Z}|U}$ is the one associated with this definition. Encoding $Z$, the lattice representation of $D$, using the code implied by $P_{\hat{Z}|U}$ incurs on the cost

$$\log \frac{1}{P_{\hat{Z}|U}(Z|U)}$$

Continuing our analysis of the optimization objective in Equation (3), the designer specified loss function value is

$$\ell(D, g(Z - U))$$

so that the total objective function is transformed to

$$\ell(D, g(Z - U)) + \lambda \log \frac{1}{P_{\hat{Z}|U}(Z|U)} \tag{8}$$

The difficulty in optimizing this objective function is that $Z$ is the result of quantization operation; this affects both terms. To deal with the term in the left, we observe that Lemma 1 implies that

$$P_{X, Z-U} = P_{X, X-U}$$

Now observe that $(Z - U \perp\!\!\!\perp D|X)$ and $(X - U \perp\!\!\!\perp D|X)$; and as a consequence

$$P_{D, Z-U} = P_{D, X-U}$$

as well. This means that

$$E_{D, Z-U}\left[\ell(D, g(Z - U))\right] = E_{D, X-U}\left[\ell(D, g(Z - U))\right] = E_{D, U}\left[\ell(D, g(e_c(D) - U))\right]$$

This demonstrates how we get rid of the quantization by replacing it with subtractive dithering for the first term in (8). For the second term (the representation cost), we use Theorem 1 to obtain

$$E_{D, U}\left[\log \frac{1}{P_{\hat{Z}|U}(Z|U)}\right] = E_{D, U}\left[\log \frac{f_U(U)}{f_{\hat{X}-U}(X - U)}\right].$$

so that the new optimization objective is now

$$E_{D, U}\left[\ell(D, g(e_c(D) - U))\right] + \lambda E_{D, U}\left[\log \frac{f_U(U)}{f_{\hat{X}-U}(f_c(D) - U)}\right]. \tag{9}$$

As indicated earlier, stochastic gradient descent is possible because $f_{\hat{X}-U}$ does not use any global information about the training data $D$. The way we propose this objective function be optimized is similar to the optimization algorithms used in Variational Autoencoders (the so called "re-parametrization trick"); in particular, we propose to sample for every $D$, one dither $U$ and then performing gradient descent regarding the sampled $U$ regarded as a constant in the computational network.

### 3.3 THE CONNECTION TO VARIATIONAL AUTOENCODERS

Assume that the role of the decoder $g$ is to output a distribution over the data space $\mathcal{D}$, this is,

$$g(Z - U) = Q(\cdot | Z - U)$$

where $Q$ is a distribution over $\mathcal{D}$. Assume that that we choose for the loss function

$$\ell(D, Q(\cdot | Z - U)) = -\log Q(D | Z - U).$$

In this setting, the loss function is quantifying the *likelihood* of the data $d$ given the dithered lattice representation $Z - U$. From an information theoretic coding perspective, this likelihood is the cost in bits needed to represent the data $d$, given $Z - U$. In information theory, $U$ is regarded as *common randomness* shared between the encoder and decoder, and therefore there is no cost in transmitting it (in practice, pseudo random numbers with a shared seed can be used to implement this, for example). Thus to complete the cost needed to represent the data $D$, we need the cost of representing $Z$, which is in our work is given by $-\log P_{\hat{Z}|U}(Z|U)$. As per our earlier development, we can substitute both of these likelihoods instead with ones which do not employ quantization, arriving to

$$\log \frac{1}{Q(D | e_c(D) - U)} + \lambda \log \frac{f_U(U)}{f_{\hat{X} - U}(e_c(D) - U)} \tag{10}$$

Those versed in the literature of variational autoencoders, might recognize the famous Evidence Lower BOund (ELBO) (for $\lambda = 1$) in the expression above. The term in the right-hand side is associated with the KL term (prior to taking expectation). One way to see is by noticing that $f_U(U) = f_{\hat{X} - U|\hat{X}}(e_c(D) - U | e_c(D))$ which is the "approximate posterior" and $f_{\hat{X} - U}$ is the "prior" (which explains why we are giving the designer full control over the distribution of $\hat{X}$). The term on the left, on the other hand, is associated with the "decoder" in the VAE.

#### 3.3.1 THE CONNECTION TO GAUSSIAN VAES

A very common practice in VAEs is to model the prior and the approximate posterior as Gaussians. It turns out that there is a sense in which the distribution of $U$ approximates that of independently identically distributed (i.i.d.) Gaussian noise. To see this, we need to introduce additional concepts from the theory of lattices.

The *volume*, *second moment* and and *normalized second moment* of the lattice cell are defined as

$$V_\Lambda = \int_{\mathcal{P}_0(\Lambda)} du, \qquad \sigma_\Lambda^2 = \frac{1}{V_\Lambda} \frac{1}{m} \int_{\mathcal{P}_0(\Lambda)} \|u\|_2^2 du, \qquad G(\Lambda) = \frac{\sigma_\Lambda^2}{V_\Lambda^{2/m}}$$

respectively. The smallest possible second moment for any lattice on dimension $m$ is defined as $G_m$; we shall call any such lattice *optimal*. The normalized second moment of an $m$-dimensional hyper-sphere is denoted by $G_m^*$. It is known that

$$G_m > G_m^* > \frac{1}{2\pi e}, \qquad \lim_{m \to \infty} G_m^* = \frac{1}{2\pi e}$$

A classical result in the theory of lattices is that the normalized second moment of an $m$ dimensional optimal lattice converges to the same quantity

$$\lim_{m \to \infty} G_m = \frac{1}{2\pi e}; \tag{11}$$

in other words, there exist lattices whose normalized second order moment approaches that of a hyper-sphere as the lattice dimension $m$ grows to infinity. Now denote $Z^*$ to be an $m$ dimensional Gaussian vector with independent entries each with a variance equal to $\sigma_\Lambda^2$, and let $U_m$ denote a random vector uniformly distributed over $\mathcal{P}_0(\Lambda)$. Then, it is not difficult to show that (Zamir et al., 2014)

$$D_{KL}(U_m \| Z^*) = \frac{1}{2} \log(2\pi e G(\Lambda)) \tag{12}$$

where $D_{KL}$ denotes the KL divergence and as a consequence, combining (11) and (13), we see that

$$\lim_{m \to \infty} D_{KL}(U_m \| Z^*) = 0. \tag{13}$$

One way to interpret this colloquially is that the distribution of the dither of a good lattice approximates that of a Gaussian as the dimension grows to infinity. This establishes more firmly the connection between variational autoencoders that employ Gaussian approximate posteriors with lattice representations.

Suppose for now that $\hat{X}$ is chosen also to be i.i.d. Gaussian. In light of (13), let us analyze the performance of a good high dimensional lattice $\Lambda$ by assuming $U$ to be also i.i.d. Gaussian with variance $\sigma_\Lambda^2$. Then $f_{\hat{X}-U}$ is the distribution of another i.i.d. Gaussian. Let us assume that the parameters of $\hat{X}$ and $U$ are such that $\hat{X} - U$ has unit variance then (15) becomes

$$\log \frac{1}{Q(D|e_c(D) - U)} + \lambda \frac{1}{2} \left( \|e_c(D) - U\|_2^2 - \frac{1}{\sigma_\Lambda^2} \|U\|_2^2 - m \log \sigma_\Lambda^2 \right) \tag{14}$$

At this point, the expression is *identical* to one that one might derive from VAE theory. One important point is to note that in VAEs, the approximate posterior can have general parameters that depend on the encoder output in complex forms; for example, in the case of a Gaussian approximate posterior both the mean and the correlation matrix in the approximate posterior could depend on the encoder output. In the case in which we use lattice representations, only the mean of the approximate posterior can have dependencies on the encoder output.

The significance of the observation in this subsection is that many of the results (with relatively minor modifications) that the VAE community has obtained likely can be re-interpreted as results involving (discrete) lattice quantized representations together with pre/post quantized dithering.

The reader may be wondering: so what is new? The point of this section is to suggest that very good lattices approximate the behavior of Gaussian VAEs which are known to be good. To implement a practical system, Theorem 1 can be used to implement finite lattices. We will show this in the experimental section.

The reader may also be asking: but why even use lattices at all? Why not use, say, categorical variables and some variant of the Concrete/Gumbel-Softmax trick, or use VQ-VAEs directly? One way to answer this question is: consider using lattices if you interested in a theoretically sound basis obtaining Gaussian VAE-like performance using discrete representations.

## 4 EXPERIMENTAL RESULTS

For our experiments with finite dimensional lattices, we will be using a lattice whose basis is a diagonal matrix with positive entries. This is a simple variant of a cubic lattice in which the spacing parameter can vary per dimension so we will call it the *rectangular* lattice. To implement training of a VAE using this lattice, we rely on Theorem 1. For this, we need to specify a prior distribution on $\hat{X}$. Note that one of the key things we will need is $f_{\hat{X}-U}$, the density of $\hat{X} - U$ where $U$ is uniformly distributed over the cell of the lattice we are considering here. Therefore, it is highly advantageous to choose a distribution for $\hat{X}$ so that the distribution of $\hat{X} - U$ has a simple form.

Our choice for the distribution of $\hat{X}$ is for each entry of the vector $\hat{X}$ to be a zero mean Laplacian distribution, this is, an origin centered symmetric distribution that decays exponentially on each side. To simplify, assume $m = 1$ and let $\Delta > 0$ the width of the uniform distribution so that $U$ is uniformly distributed over $(-\Delta/2, \Delta/2)$. Let $b$ be the parameter for the Laplacian distribution of $\hat{X}$, then

$$f_{\hat{X}-U}(\eta) = \begin{cases} 1 - \frac{1}{2} \exp\left(-\Delta/2b\right) \left(\exp(-\eta/b) + \exp(\eta/b)\right) & |\eta| < \Delta/2 \\ \frac{1}{2} \exp(-|z|/b)) (\exp(\Delta/2b) - \exp(-\Delta/2b)) & |\eta| \geq \Delta/2 \end{cases}$$

In general we assume a collection of $\{\Delta_i\}$ and $\{b_i\}$ potentially distinct parameters.

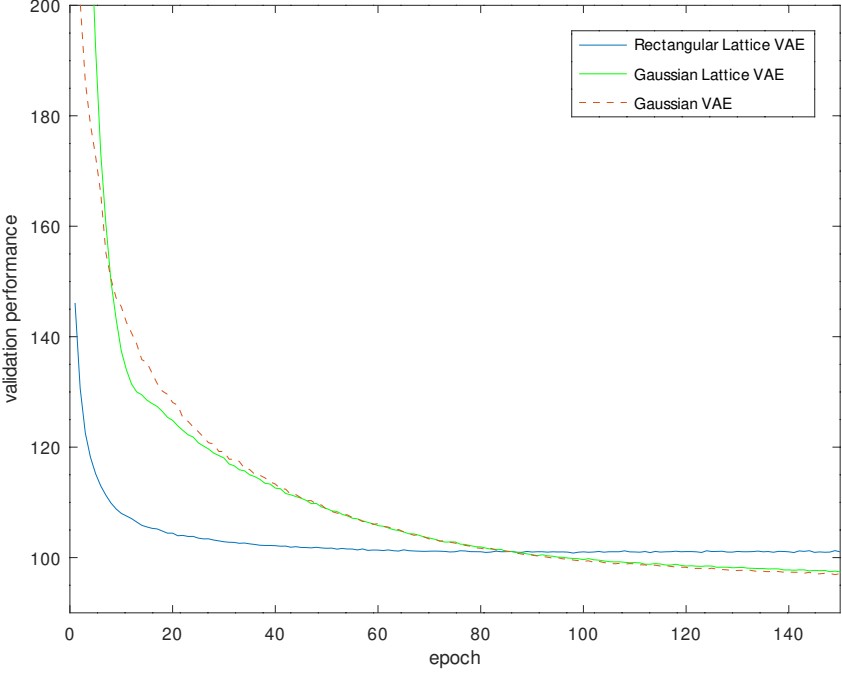

Figure 3: Comparison of the performance (negative log likelihood in nats) of a rectangular lattice VAE, a hypothetical high dimensional lattice (in green), approximated using a Gaussian VAE where the variance of the approximate posterior is not data dependent, and and a Gaussian VAE (in red) where the variance of the Gaussian of the approximate posterior is allowed to be data dependent.

We use static MNIST in our experiments. For both the encoder we use a three layer network in which the dimensions of the intermediate layer outputs are $784 \rightarrow 300 \rightarrow 300 \rightarrow 40$. The first two layers are gated linear networks with no activation function (the only nonlinearity is the multiplicative gating) and the last layer is a simple linear network with no gating or activation. For the decoder the dimensions are $40 \rightarrow 300 \rightarrow 300 \rightarrow 784$; as before the first two layers are gated linear networks with no activation function but the last layer is an ungated linear network with a sigmoid activation function. We use a batch size of 100 and Adam with learning rate 5e-4. We keep track of the model with the best validation set performance and if more than 50 epochs pass without being able to improve this performance, we stop the experiment. Then we compute the an improved log likelihood estimate by making use of importance sampling with 1000 samples for each test example Burda et al. (2016). To implement this, we adapted the code made available for the results in Tomczak & Welling (2018).

When training for the finite dimensional lattice, the training objective for an MNIST image $D$ is

$$\log \frac{1}{Q(D|e_c(D) - U)} + \sum_{i=1}^{m} \log \frac{1}{\Delta_i[f_{\hat{X}-U}(e_c(D) - U)]_i} \tag{15}$$

where $Q$ is the decoder network, $e_c$ denotes the encoder network and $[]_i$ denotes the $i$th element of a vector.

When train two types of Gaussian VAEs. In one case, the approximate posterior variance is not allowed to depend on the original MNIST image; we call this the Lattice Gaussian VAE because it denotes the performance of an ideal lattice VAE. The other setting is a Gaussian VAE where the approximate posterior variance is allowed to depend on the data; this we simply term Gaussian VAE.

Table 1: Comparison of test negative log likelihood using importance sampling (nats)

| Rectangular lattice | Gaussian lattice VAE | Gaussian VAE |
|---|---|---|
| 94.45 | 89.35 | 88.85 |

The results of our experiments can be seen in Figure 3, where we illustrate the negative log likelihood of the validation set as a function of the number of training epochs. As seen, the performance of the rectangular lattice VAE is not as good as that of either of the two Gaussian VAEs, which is expected; as described in this article, simpler lattices won't perform as well as high dimensional good lattices. Notice however that the performance of the rectangular lattice is quite competitive and furthermore, notice that the performance of the two Gaussian VAEs are rather close to each other. This is encouraging for lattice based VAEs because it implies that it is in principle possible to get rather close to the performance of a Gaussian VAE using higher dimensional lattices.

We then evaluated a better upper bound on negative log likelihood through the technique of importance sampling (Burda et al., 2016); the results are in Table 1. We see here that the performance of the rectangular lattice VAE remains quite competitive; for example the numbers reported for Gumbel-Softmax for MNIST in Jang et al. (2017) are above 100 nats.

To get closer to the performance of a Gaussian Lattice VAE, what we need to do is to increase the effective lattice dimension; for example we could use the hexagonal lattice or even higher dimensional lattices. Theorem 1 remains the main tool for training such more complex systems; this is left for future work.

## 5 CONCLUSIONS

The present work is inspired by a belief that information theory, and in particular lossy compression theory can be very effective in serving as a theoretical foundation for problems in representation learning, including the design and analysis of highly performant practical algorithms. We have introduced lattices as a possible way to create discrete representations, and proved a fundamental result which allows us to train computational networks that use lattice quantized dithering using an equivalent (in an expected sense) computational network which replaces quantization with dithering, thus allowing gradient descent to apply. This result also allows us to use only local information during the optimization, thus additionally enabling *stochastic* gradient descent. We also established a fundamental connection between the use of good high dimensional lattices and the idea of Gaussian dithering, which is common in generative modeling settings such as Variational Autoencoders. Finally, we provided initial experimental evidence of the potential of using lattices in an VAE setting, where we contrasted the performance of a rectangular lattice based VAE and two types of Gaussian VAEs. The bottom line is that if one is interested in getting close to the performance of a Gaussian VAE with discrete representations with a good theoretical basis, we suggest the reader to consider lattices and to train them using dithered stochastic gradient descent.

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

## A  APPENDIX

### A.1  PROOF OF THEOREM 1 - CORE RESULT ON DITHERED SGD

For reference purposes, we restate the statement of the theorem:

**Theorem 1** (representation cost for dithered SGD). *Let $\Lambda$ be any lattice. Let $X$ be a random $\mathbb{R}^m$ vector distributed according to $P_X$, $\hat{X}$ be a random $\mathbb{R}^m$ vector distributed according to $P_{\hat{X}}$, where we assume that $P_X \ll P_{\hat{X}}$. Assume $U$ is uniformly distributed over $\mathcal{P}_0$, and assume that $U \perp\!\!\!\perp X$ and $U \perp\!\!\!\perp \hat{X}$. Define*

$$Z = K_\Lambda(X + U) \tag{16}$$
$$\hat{Z} = K_\Lambda(\hat{X} + U) \tag{17}$$

*Then*

$$E_{X,U}\left[\log\frac{1}{P_{\hat{Z}|U}(Z|U)}\right] = E_{X,U}\left[\log\frac{f_U(U)}{f_{\hat{X}-U}(X-U)}\right]. \tag{18}$$

To prove this result, we will rely on this Lemma:

**Lemma 2.** *Let $X, U, Z$ be $\mathbb{R}^m$ valued random vectors such that $(X \perp\!\!\!\perp U|Z - U)$ and $X \perp\!\!\!\perp U$, with $U$ continuous and $Z$ discrete. Then*

$$\frac{P_{Z|XU}(z|xu)}{P_{Z|U}(z|u)} = \frac{f_{Z-U|X}(z-u|x)}{f_{Z-U}(z-u)}$$

**Proof of Lemma 2.** We assume in this proof that $X$ is continuous and has a density; the case where $X$ is discrete can be proved similarly. The proof proceeds as follows:

$$
\begin{aligned}
\frac{P_{Z|XU}(z|xu)}{P_{Z|U}(z|u)} &\overset{(a)}{=} \frac{f_{X|Z,U}(x|z,u)}{f_{X|U}(x|u)} \\
&\overset{(b)}{=} \frac{f_{X|Z,U}(x|z,u)}{f_X(x)} \\
&\overset{(c)}{=} \frac{f_{X|Z-U,U}(x|z-u,u)}{f_X(x)} \\
&\overset{(d)}{=} \frac{f_{X|Z-U}(x|z-u)}{f_X(x)} \\
&= \frac{f_{X|Z-U}(x|z-u)f_{Z-U}(z-u)}{f_X(x)f_{Z-U}(z-u)} \\
&\overset{(e)}{=} \frac{f_{Z-U|X}(z-u|x)}{f_{Z-U}(z-u)}
\end{aligned}
$$

where $(a)$ follows from Bayes' rule for mixed continuous/discrete variables, $(b)$ follows from the assumption that $X \perp\!\!\!\perp U$, $(c)$ follows from the fact that these two events are identical

$$
[Z = z, U = u] = [Z - U = z - u, U = u].
$$

Finally, $(d)$ follows from the assumption that $(X \perp\!\!\!\perp U | Z - U)$ and $(e)$ follows from the use of Bayes' rule.

$\square$

**Proof of Theorem 1.** A fundamental fact in probability theory is that if $A$ and $B$ are random vectors where $B = \eta(C)$ for some function $\eta$, then $A$ and $B$ are independent given $C$. We now use the classical argument employed in the analysis of lattices in quantization, which is that because by definition $U \in \mathcal{P}_0(\Lambda)$,

$$
K_\Lambda(\hat{Z} - U) = \hat{Z}
$$

and therefore $K_\Lambda(\hat{Z} - U) - (\hat{Z} - U) = U$. This shows that $U$ can be obtained as function of $\hat{Z} - U$. Therefore $(\hat{X} \perp\!\!\!\perp U | \hat{Z} - U)$ and since by construction $\hat{X} \perp\!\!\!\perp U$, we will be able to apply Lemma 2.

The proof thus proceeds as follows:

$$
E_{X,U}\left[\log\frac{1}{P_{\hat{Z}|U}(Z|U)}\right] \stackrel{(a)}{=} E_{X,U}\left[\log\frac{P_{\hat{Z}|U,\hat{X}}(K(X+U)|U,X)}{P_{\hat{Z}|U}(Z|U)}\right]
$$

$$
\stackrel{(b)}{=} E_{X,U}\left[\log\frac{P_{\hat{Z}|U,\hat{X}}(Z|U,X)}{P_{\hat{Z}|U}(Z|U)}\right]
$$

$$
\stackrel{(c)}{=} E_{X,U}\left[\log\frac{f_{\hat{Z}-U|\hat{X}}(Z-U|X)}{f_{\hat{Z}-U}(Z-U)}\right]
$$

$$
\stackrel{(d)}{=} E_{X,Z-U}\left[\log\frac{f_{\hat{Z}-U|\hat{X}}(Z-U|X)}{f_{\hat{Z}-U}(Z-U)}\right]
$$

$$
\stackrel{(e)}{=} E_{X,X-U}\left[\log\frac{f_{\hat{Z}-U|\hat{X}}(X-U|X)}{f_{\hat{Z}-U}(X-U)}\right]
$$

$$
\stackrel{(f)}{=} E_{X,U}\left[\log\frac{f_{\hat{Z}-U|\hat{X}}(X-U|X)}{f_{\hat{Z}-U}(X-U)}\right]
$$

$$
\stackrel{(g)}{=} E_{X,U}\left[\log\frac{f_{\hat{X}-U|\hat{X}}(X-U|X)}{f_{\hat{X}-U}(X-U)}\right]
$$

$$
= E_{X,U}\left[\log\frac{f_U(U)}{f_{\hat{X}-U}(X-U)}\right]
$$

where $(a)$ follows from the definition of $\hat{Z}$ which implies that for all $x$ and all $u$,

$$
P_{\hat{Z}|U,\hat{X}}(K_\Lambda(x+u)|u,x) = 1,
$$

$(b)$ follows from the definition of $Z$, $(c)$ follows from an application of Lemma 2, $(d)$ is a change of variables which is feasible since $X$ and $Z-U$ are the only random vectors under the expectation, $(e)$ follows from an application of the Crypto Lemma, $(f)$ is another change of variables, $(g)$ follows from another application of the Crypto Lemma, but this time applied to the joint distribution of $\hat{X}, \hat{Z}-U$. $\qquad\square$

## A.2 Additional results on an autoencoding setting

In this appendix, we describe some results for the autoencoding setting, where the goal is not to estimate densities but rather to create a representation that describes with high fidelity the original object of interest.

In the autoencoding setting we are able to make a stronger statement than in the VAE example in the main body of the paper about the benefits of using better lattices.

Let $I_m$ denote the $m \times m$ identity matrix. The following Lemma describes how the second moment of a lattice whose dither is "white" controls the quantization error as observed through an arbitrary function:

**Lemma 3** (approximation error due to lattice quantization). *For a given dimension $m \geq 1$, let $\eta : \mathbb{R}^m \to R$ be a twice differentiable function. Assume that $U$ is a column vector uniformly distributed over the cell $\mathcal{P}_0(\Lambda)$ of a lattice $\Lambda$, and assume that $E_U[UU^T] = \sigma_\Lambda^2 I_m$ for some $\sigma_\Lambda^2$. If $tr(\mathbf{H}_\eta(x)) \neq 0$ and if $\sigma_\Lambda^2$ is sufficiently small, then*

$$
E_U[\eta(x+U)] - \eta(x) \approx \frac{\sigma_\Lambda^2}{2}\mathrm{Tr}(\mathbf{H}_\eta(x))
$$

*where $\mathbf{H}_\eta$ denotes the Hessian (the matrix of second order derivatives) of $\eta$.*

**Proof.** It is easy to see that if $u \in \mathcal{P}_0(\Lambda)$, then $-u \in \mathcal{P}_0(\Lambda)$. Due to this property, the first moment of a lattice is zero:

$$
E_U[U] = 0.
$$

Let $\Lambda_{scaled} = \frac{1}{\sigma_\Lambda}\Lambda$ be a *scaled* version of $\Lambda$. Define $V = \frac{1}{\sigma_\Lambda}U$, then clearly $E_V[VV^T] = I_m$. Using a Taylor series expansion, assuming $\sigma_\Lambda$ is small,

$$
\begin{aligned}
E_U[\eta(x+U)] = E_V[\eta(x+\sigma_\Lambda V)] &\approx \eta(x) + E_V\left[\frac{\sigma_\Lambda^2}{2}V^T\mathbf{H}_\eta V\right] \\
&= \eta(x) + \frac{\sigma_\Lambda^2}{2}\left(\sum_{i,j}[\mathbf{H}_\eta]_{i,j}E_V[V_iV_j]\right) \\
&= \eta(x) + \frac{\sigma_\Lambda^2}{2}\operatorname{Tr}(\mathbf{H}_\eta).
\end{aligned}
$$

$\square$

Lattices that achieve $G_m$ have white dithers. This is a result by Poltyrev and can be found in (Zamir & Feder, 1996):

**Lemma 4.** *For any $m \geq 1$, if a lattice $\Lambda$ that achieves $G_m$ and $U$ is uniformly distributed over $\mathcal{P}_0(\Lambda)$, then $E[UU^T]$ is proportional to $I_m$.*

If one keeps constant the volume $V_\Lambda^{2/m}$ of the lattice cell then by definition,

$$
\sigma_\Lambda^2 = V_\Lambda^{2/m}G(\Lambda) \geq \frac{V_\Lambda^{2/m}}{2\pi e}
$$

We consider two lattices: a lattice spanned by a matrix proportional to $I_m$, and a lattice with a normalized second moment equal to $G_m$. From the previous discussion, it is easy to see that both lattices have white dithers, and therefore Lemma 3 applies. We also note that the normalized second moment of lattice spanned by a matrix proportional to $I_m$ is $G_1 = 1/12$. Applying Lemma 3 twice, the difference between the two approximations (one for each lattice) can be estimated by

$$
\frac{1}{2}(G_1 - G_m)|\operatorname{Tr}(\mathbf{H}_\eta(x))| \leq \frac{1}{2}\left(\frac{1}{12} - \frac{1}{2\pi e}\right)|\operatorname{Tr}(\mathbf{H}_\eta(x))| \approx 0.0124|\operatorname{Tr}(\mathbf{H}_\eta(x))|
$$

The above is an absolute estimate. In relative terms, the size of the error by using the best lattice with dimension $m$ relative to the size of the error incurred by the cubic lattice. This is then given by

$$
\frac{G_m}{G_1} \geq \frac{6}{\pi e} \approx 0.7
$$

In the above, $\eta$ was a generic function but in our case, we would like to identify it with $\ell$. Thus, we can improve the quantization error in the designer's choice of a loss function by up to 30% by using a better lattice, all at a constant representation cost.

### A.2.1 AUTOENCODING EXPERIMENTS

The purpose of our experiments is to draw a comparison between two techniques for quantized representation learning. Both techniques will be using stochastic gradient descent, which implies that we need to guarantee that the "backwards" computational network (the one used to compute the gradient of the objective functional with respect to all parameters) needs to be associated with a meaningful gradient, this is, one that can be used to iteratively improve the loss function.

The first technique is a network trained with scalar or vector quantization in the forward direction. Because quantization has a zero derivative almost everywhere, it is not "meaningful" in the sense of the paragraph above. Therefore, in the backward direction we use straight-through estimation in order to create a fully differentiable backwards computational network (Bengio et al., 2013; van den Oord et al., 2017). The second technique is an estimate of how a very good lattice in high dimensions will perform, using non-data dependent Gaussian dithering to approximate a uniform distribution over this lattice.

For the experimental setting, we have chosen a `seq2seq` (Sutskever et al., 2014) autoencoder as implemented in `OpenNMT-py` (Klein et al., 2017; 2018). This open source package implements a generic `seq2seq` architecture that encodes a variety of input types (text, audio, images) into a

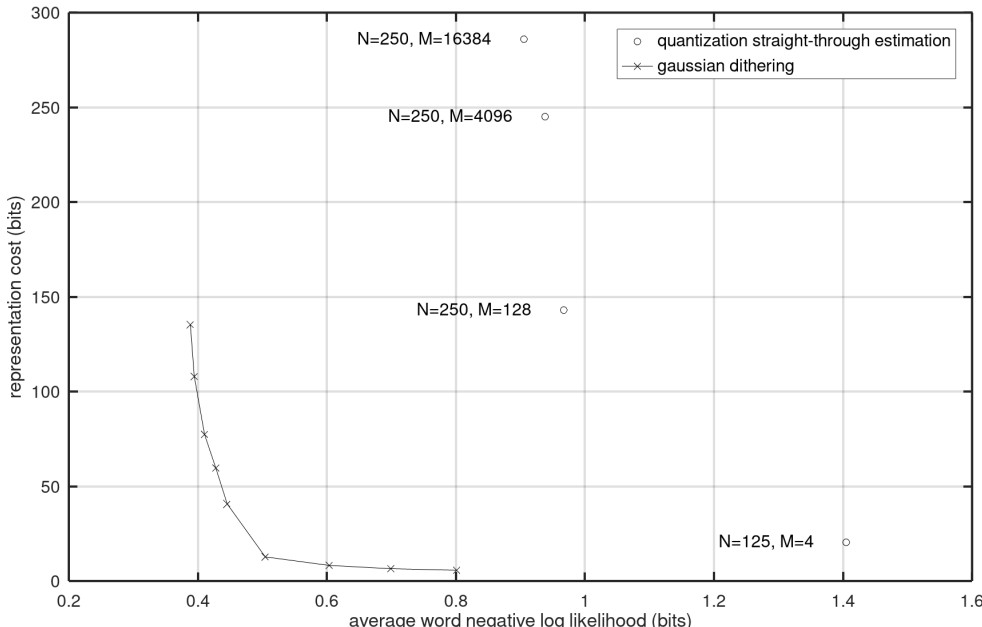

Figure 4: Comparison of the performance of a hypothetical high dimensional lattice, approximated using Gaussian dithering, and several quantizers trained using straight-through estimation. The x-axis is the average word negative log likelihood in bits. The y-axis is the representation cost of the quantized representation vector, averaged per word. Lower is better for both. All numbers plotted are for the *test* data.

vector valued representation, and then decodes this representation into text. For our experiments, we have chosen a strict autoencoder experiment where we encode text into a representation and where we expect that exactly the same text be produced by the decoder from this representation.

We modified this open source package (the source code will be released) by intercepting the communication between the encoder and decoder in order to apply the various quantization techniques that we are proposing to evaluate. In a `seq2seq` setup like the one being described in here, the *loss function* is the average word negative log likelihood at the output of the decoder. We will maintain the same loss function verbatim. In addition, we computed the representation cost of the quantized vector communicated from the encoder to the decoder, normalized also on a per word basis. There is a tradeoff between these two quantities - the higher the allowed representation cost, in principle the better loss function we can obtain.

For the specifics of the architecture, we chose a 2 layer bidirectional GRU (Cho et al., 2014) recurrent neural network for both the encoder and decoder, with each direction having 250 dimensions in its internal state and in the GRU's output vector (so that the total communication pipe between encoder and decoder is 500 dimensions). The `OpenNMT-py` package implements an optional *global attention* which creates additional communication paths between encoder and decoder. We disabled this, as it presents additional complications that are not important for the purposes of this article. We use Adam with an initial learning rate of 1e-4, which decays by a factor of 0.75 every 50k steps after an initial 100k steps where it is kept constant. The total number of training steps is 500k. The parameter $\lambda$ which weights the representation cost in the total objective function is slowly annealed from a value close to 0 to its (asymptotic) value of 1.0; the value of 0.5 is achieved at 100k steps.

When performing quantization and straight-through estimation, the 250 dimensions of each direction of the bi-directional GRU can be quantized in several ways, depending on how we partition the 250 dimensions. One family of such possibilities is to use $N = 250$ scalar quantizers each using $M$ levels, for various values of $M$. Another example we demonstrate is $N = 125$ quantizers, each with $M = 4$ two dimensional code vectors. In any of the experiments, each quantizer is independent of

the other quantizer, in the sense that its $M$ levels are parameters of the model which are optimized using gradient descent. Each of the $N$ codes is associated with a *code* (a distribution over its $M$ levels), which can also be optimized using gradient descent. The average code length obtain as the representations are encoded with this code is the *representation cost*.

For the experiments using Gaussian dithering, the output of the encoder is first linearly transformed using a linear operator with free parameters, and then it is dithered using an uncorrelated zero mean Gaussian with a diagonal correlation matrix that is also a set of free parameters. The dithered signal is then passed through another linear operator. The representation cost for Gaussian dithering is computed using techniques from Variational Autoencoders; in essence we assume an isotropic unit variance Gaussian as a prior over the representation space and then estimate the KL divergence between the dithered representation conditional on the encoder output and the Gaussian prior.

The specific data that we use for the text autoencoding experiment is a set of over 4 million english sentences prepared using scripts that are available in Google's NMT tutorial (`https://github.com/tensorflow/nmt`, in particular the `wmt16_en_de.sh` script) for a German/English translation system; as this is an autoencoding setup, we only use the English sentences. For the test data, we use 3003 sentences also typically used in translation experiments for testing (`newstest2014`). This data set was then processed using `OpenNMT-py preprocess.py` tool. We only report results for the test data.

The results can be seen in Figure 4. In the lower left hand side we see the tradeoff between the representation cost and the average word negative log likelihood. We can see that the projected performance of a good lattice is significantly better than the performance of the specific quantizers, trained with straight-through estimation that we tested. The reader may be wondering whether the high dimensional assumption that we make on the gaussian dithering approximation to good lattices implies that the projected performance may be unrealistic; we do not know with certainty at this point however we believe that a good lattice in 250 dimensions will likely be very well approximated with a Gaussian.

