# OpenReview forum: "Lattice Representation Learning"
_ICLR.cc/2020/Conference — Reject_

### Official Review · AnonReviewer3 · 2019-10-21
**Official Blind Review #3**

**Rating:** 3

**Review:**


The paper discusses discrete representation learning from a lattice perspective, where one performs a "reparametrization trick" during training and quantized learning over inference. This is different from other methods such as Gumbel trick in the sense that the quantization is done on lattices and the noise is uniform over primitive cells. The paper discusses some connections with VAEs, showing how one could interpret the VAE objective with "Gaussian dithering". While the theory is interesting and seems to be valid, I am not entirely sure how we could use it well in practice (apart from the proposed Gaussian dithering, which is very similar to what we already have with VAE reparametrization trick).


Questions:
	- In sec 2.3 an argument is placed for connection with VAEs. It seems that here U is sampled from Gaussian and therefore unbounded and not "uniform" within the primitive cell? It's not critical, but I wonder if the this argument could be more interested if better connected to lattices, ie the optimality of "hyperspheres" as a lattice representation.

	- Experimental-wise, I don't see how the proposed training approach is different from a regular VAE setup (maybe that is the point), and I am confused by the procedure in which you would do quantization for the "hypothetical lattice" experiments. Does the "representation cost" here mean the loss you get with the Gaussians? How do we obtain useful discrete representations in practice?

	- It seems helpful to have experiments on popular tasks for VAEs like binarized MNIST; this would make it easier to compare with more existing baselines.

	- It might be nice to restructure the paper, a lot of the paragraphs are very long and a lot of math uses one entire line while being very short (it seems the numbered equations are most useful). Some details can be put in the appendix to make the paper shorter.

Typos:
	- introduction "a line of research of autoencoders" \citep format.
	- Description for (3) "first term" and "second term" order?
	- Lemma 1: "a uniformly distributed over"
	- Grammatical errors across the paper; it would be nice to proof read them carefully for camera-ready version.



**Experience Assessment:**

I have read many papers in this area.

**Review Assessment: Checking Correctness Of Derivations And Theory:**

I assessed the sensibility of the derivations and theory.

**Review Assessment: Checking Correctness Of Experiments:**

I carefully checked the experiments.

**Review Assessment: Thoroughness In Paper Reading:**

I read the paper at least twice and used my best judgement in assessing the paper.

---

> ### Author Response · Authors · 2019-11-15
> **Appreciate the comments - they motivated quite a bit the changes in the new version**
>
>
> A main point of Reviewer 3 is that "While the theory is interesting and seems to be valid, I am not entirely sure how we could use it well in practice (apart from the proposed Gaussian dithering, which is very similar to what we already have with VAE reparametrization trick).". We agree that  could be used wasn't immediately obvious from the way we wrote the paper. We believe that this is because we didn't include a specific description of how it is that one would use finite lattices nor included experimental results on finite lattices. The new version of the paper now includes that.
>
> On reviewer 3's question " In sec 2.3 an argument is placed for connection with VAEs....". There was a similar concern from Reviewer 4. In the new version of the paper, we make it clear that uniform distributions of cells of good lattices resemble those of Gaussian distributions in a specific sense (convergence as measured by KL divergence).
>
> - "Experimental-wise, I don't see how the proposed training approach is different from a regular VAE setup (maybe that is the point)". Correct, this is the essential point. As a result of this work,  we now know that we can use regular continuous VAE ideas in training, and then use Theorem 1 during inference time to create dithered lattice representations, which are discrete conditional on the dither.  When working on finite dimensions, we do not Gaussian random variables, but many of the traditional ideas on VAEs can be use with some adaptation. We now show a specific example involving a cubic lattice to make this concrete.
>
> - "How do we obtain useful discrete representations in practice?" Through Theorem 1 you can obtain discrete representations conditional on the dither value. Once you've trained your network, you use the encoder, add a dither, and then quantize using your choice of lattice (in our experiments, a cubic lattice). That representation is discrete. Before using the representation, you subtract the dither, and feed to the decoder.
>
> - "It seems helpful to have experiments on popular tasks for VAEs like binarized MNIST; this would make it easier to compare with more existing baselines. " Absolutely. This comment led to us replacing our existing experimental section with a simpler one, with benefits in both clarity and reproducibility. Thank you very much for encouraging us to do this.
>
> We also fixed the typos found by Reviewer 3. Thank you very much!!

---

### Official Review · AnonReviewer1 · 2019-10-27
**Official Blind Review #1**

**Rating:** 3

**Review:**

**After rebuttal**

Thanks for the work you put into the rebuttal!

I think the paper now reads better and like the new added experiment! However, I remain unconvinced about the practical usefulness of lattice representations especially in light of missing comparison to other algorithms (as mentioned, I am no expert, but [1] or something similar would have been nice). I will thus leave the “weak reject” score unchanged but will not block the acceptance if other reviewers and/or the AC believe the paper should be accepted.


**Original review**

This paper presents a technique for learning of lattice valued embeddings. The authors propose to learn the embeddings by gradient descent which naturally brings about the challenge of how to obtain non-zero gradients when the embedding space is discrete. To resolve this, they apply the “Crypto-Lemma”, a well-known result from information theory, which allows them to obtain meaningful gradients by using certain uniform perturbation of the embeddings. Since the resulting algorithm is hard to implement in high dimensions, the authors eventually replace the uniform noise with a small Gaussian noise. The paper is concluded with an experiment comparing learned and fixed lattice embeddings in terms of the average code length (“representation cost”) on a dataset of English sentences.

I am currently leaning towards recommending rejection of this paper. The two main reasons are: (i) I am missing concrete examples of where application of lattice embeddings is more beneficial than simple continuous (or binary) valued embeddings; (ii) If the main application of this algorithm is supposed to be compression, then I am missing sufficient background section on alternatives to the presented algorithm and comparison to these within the experiments (I am admittedly no expert on the use of ML algorithms for compression, but a simple search reveals, e.g., [1] as a relevant baseline). Finally, since the paper is 10 pages, I am applying a higher standard then I would to an 8 page paper as instructed by the guidelines.


Major comments:

- As alluded above, can you please clarify whether the main application of your algorithm is compression, or whether there are applications in which lattices have interpretative value or some other advantage not related to compression?

- Can you please explain why is there no comparison to other compression algorithms (e.g., [1])?

- Can you please provide more detail on how exactly you obtain the measurements in fig.3? E.g., how have you selected the reported hyper-parameters? Were the same hyper-parameters used for both yours and the straight-through model, or did each model run with its optimal hyper-parameters? Are the reported numbers an average over a larger number of random seeds? etc.


Minor comments:

- On p.1, line 5, “of of” -> “of”

- On p.1, wasn’t the phrase “in another line of thinking” supposed to be connected to the following, instead of appended to the preceding sentence?

- On p.3, you say “We are interested in a machine learning application, but want to emphasize the representation cost for the objects being encoded as a first class metric to be optimized for.”
I am not entirely sure I understand what you want to say here. Standard variational inference (the type that is among else being employed by the cited VAEs) has a well known Minimum Description Length (MDL) interpretation. Can you please clarify why MDL is not “a first class metric to be optimized for”?

- At times I felt like the paper is burdening the reader with unnecessary definitions. For example, why does the reader have to know what a “fundamental Voronoi cell” is when they already know the definition of a “fundamental cell”?

- I like that you are distinguishing between probability distributions and their density/probability mass functions (PMF) as it generally makes reader’s life easier in a paper like this. However, I would have liked if you have consistently stuck with the notation that you introduce (f for densities, p for probability mass functions) instead of using capital P in some places where density/PMF is appropriate (for example, compare the lower case p in eq.3, with the expressions for entropy and cross-entropy on p.5). On a related note, I find the choice of denoting both the encoder and density functions by f somewhat unfortunate.

- In the second display after eq.8, should S be X?


References:

[1] James Townsend, Tom Bird, David Barber. Practical Lossless Compression with Latent Variables using Bits Back Coding. https://arxiv.org/abs/1901.04866

**Experience Assessment:**

I do not know much about this area.

**Review Assessment: Checking Correctness Of Derivations And Theory:**

I assessed the sensibility of the derivations and theory.

**Review Assessment: Checking Correctness Of Experiments:**

I assessed the sensibility of the experiments.

**Review Assessment: Thoroughness In Paper Reading:**

I read the paper at least twice and used my best judgement in assessing the paper.

---

> ### Author Response · Authors · 2019-11-15
> **Thank you!! Adopted nearly all the suggested changes**
>
>
>
> - "(i) I am missing concrete examples of where application of lattice embeddings is more beneficial than simple continuous (or binary) valued embeddings;..."
>
> Understood. In the updated version of the paper we have 1) a simpler, better known setting (VAEs with static MNIST) and an example of a finite dimension lattice which makes the results quite practical.
>
> As a result of our paper enhancements, we the main storyline reads as follows (notice, same argument used responding to Reviewer 4): "Gaussian VAEs are really good, but not discrete, what if we could provably approximate their performance using discrete latent variable models? Here's how you do it with lattices.". This is not something that can be claimed, to the best of our knowledge, by techniques such as Gumbel-Softmax based discrete latent variable models. As far as the relation to VQ-VAEs, we now clarify in the text that lattice representations, when used in a VAE setting, can be seen as a form of a VQ-VAE in which the vector quantizers come from a structured codebook instead of being specified as free parameters in a machine learning model.
>
> - "(ii) If the main application of this algorithm is supposed to be compression...".  Compression is certainly one possible application; in general, wherever discrete representations are interesting, lattice representations are a possible candidate that can be evaluated against other possible alternatives. In the new version of the paper, in the experimental portion we present a VAE setup which is very closely associated with the problem of data compression, since the goal of the VAE is density estimation which is a key aspect of data compression.
>
> - "Can you please provide more detail on how exactly you obtain the measurements in fig.3?"
> There was no tuning/grid search of hyperparameters for neither of the models - we used default settings as they are used in OpenNMT. Also we are not reporting numbers as an average of a large number of random seeds. This is a great suggestion but it was difficult to do this as each experiment lasted for a few days and we wanted enough individual experiments to illustrate the tradeoff between representation cost and reconstruction performance.
>
> - "On p.1, wasn’t the phrase “in another line of thinking”  Fixed, thanks.
>
> - "On p.3, you say “We are interested in a machine learning application, ...".  We do emphasize that our work applies to a setting that is more generic than VAEs, which explains why we felt we had to justify the objective function without directly alluding to the VAE literature.
>
> - "At times I felt like the paper is burdening the reader ". Agreed. The specific problem that the reviewer raises has been corrected. I understand the general feeling - we felt we had to introduce various general concepts since we expected the audience not to be entirely familiar with lattices and wanted to enable the audience their own exploration into the literature of lattices if they felt encouraged to pursue further the ideas of this paper.
>
> - "I would have liked if you have consistently stuck with the notation" Point well taken!! Fixed.
> - "On a related note, I find the choice of denoting both the encoder and density functions by f somewhat unfortunate." Agreed. We've fixed this
>
> - "In the second display after eq.8, should S be X?  "  Yes!!! Very good catch.

---

### Official Review · AnonReviewer4 · 2019-10-28
**Official Blind Review #4**

**Rating:** 3

**Review:**

This paper proposes a framework for gradient descent optimization of latent variable models where the latent code lies on a (n-dimensional) lattice. After a brief overview of relevant results in lattice theory, a method ("Dithered gradient descent") is proposed to differentiate through lattice quantization by means of an additive "dither" noise variable and a lower bound on the total reconstruction+representation loss using a prior distribution. After elaborating on the relationship between their framework and the VAE literature, the authors go on to give an explanation of the link between the covering property of a lattice and the reconstruction loss. Finally, empirical evidence is given for the better compression and reconstruction properties of a sentence autoencoder with lattice latent codes.

Overall, while the idea presented in this paper is interesting (using lattices as discrete latent variables), I don't think that the paper in its current state is up to standards for ICLR.

First, there is little motivation for using lattices over other discrete latent variable models in the first place (eg. straight-through estimator with Gumbel softmax or VQ-VAE).

Second, while I appreciate the efforts that the authors have gone through to make lattice theory accessible to a profane audience, I think that the presentation (particularly in 2.2) could benefit from significant re-ordering as well as more emphasis on the parallel with the VAE literature. A lot of the concepts introduced there have equivalents in VAE lingo (prior, ELBO) and would benefit from being identified as such as they are introduced (and not just in a following sub-section). I also think that presenting Theorem 1 first is confusing, because we are shown the solution before seeing the problem. I think that it would be much more natural to 1. recall the original objective 2. introduce (and motivate) tithering and the $\hat X$ and then 3. introduce theorem 3 to explain how to get rid of quantization. In terms of global structure the paper is also rather imbalanced with 3 large sections including a monolithic 3 page long introduction. I would advise splitting up as appropriate to give the reader some space to breathe.

Third, and perhaps most importantly, the experimental section is underwhelming. The description of the method used is too sparse and ridden with inaccuracies and vague descriptions. A reader would be hard-pressed to try to reproduce these results from the paper alone. The task itself is rather arbitrary (sentence autoencoding with GRUs), and from what I understand, there is no direct comparison with the relevant literature (eg. Gumbel straight-through or VQ-VAEs). I think that the authors should have at least illustrated the appeal of lattice representation learning on a toy example. Furthermore as stated by the author in the conclusion, the experiments don't even involve actual finite dimension lattices, making most of the paper seem irrelevant in hindsight. Overall this section feels rushed and it is the biggest point against the paper for me.

Finally, the paper is littered with small typos or unfortunate notation collisions which, if individually benign, make parsing this already dense paper harder than needed. Some specific examples are given in the notes at the end of my review.

Notes (in no specific order):

- Second to last paragraph on page 3: "one may want to have $\Delta>0$ be very small [...] but that leads to a larger reconstruction cost". I assume you meant "representation cost".
- $- \log p$ instead of $\log \frac 1 {p}$ to save space
- Voronoi cells are introduced but never referenced.
- Subscripted lowercase f is overloaded, use lowercase p or q for densities. This makes some of the formulas in 2.2 particularly annoying to parse.
- In 2.4 $G_1$ refers to something different than $G_m$ with $m=1$
- The term "dither" is not defined explicitly. The closest I could find is a reference to "the dither value U.". Given the prominence of the term in the paper (and in the name of your proposed method) I think this warrants more attention.
- In the experimental section you justify the use of Gaussian dithering (instead of the uniform dithering used in all the theorems) by "The justification for using Gaussian dithers is partly contained in the high resolution analysis done in Subsection 2.4.". However there is no explicit reference to this fact in 2.4 (in fact the words "gaussian" or "normal" don't even appear in 2.4). The link should be made more explicit somewhere.
- Some sentences seemingly don't make sense: "The representation cost for Gaussian dithering using techniques from Variational Autoencoders; in essence we assume an isotropic unit variance Gaussian as a prior over the representation space and then estimate the KL divergence between the dithered representation conditional on the encoder output and the Gaussian prior." in the experimental section.
- Page 3 codelength: references (eg. the shannon paper). In general a lot of the "It is well-known" statements in the paper should be accompanied by a reference.
- Some references don't have a date (Van den Oord in page 9 for instance)
- the result figure is very hard to read especially on paper. A bit of color as well as larger font/linewidth would go a long way.



**Experience Assessment:**

I do not know much about this area.

**Review Assessment: Checking Correctness Of Derivations And Theory:**

I assessed the sensibility of the derivations and theory.

**Review Assessment: Checking Correctness Of Experiments:**

I assessed the sensibility of the experiments.

**Review Assessment: Thoroughness In Paper Reading:**

I read the paper at least twice and used my best judgement in assessing the paper.

---

> ### Author Response · Authors · 2019-11-15
> **Thank you for your thoughtful remarks**
>
>
> We agree that the experimental section can be improved. Both reviewer 4 and reviewer 3 suggest using a toy example, as this helps with understand and reproducibility. We have replaced now in the main body of the text the experiments section with a simpler set of experiments done on static MNIST.
>
> Critically, reviewer 4 points out that "... the experiments don't even involve actual finite dimension lattices, making most of the paper seem irrelevant in hindsight. ". Point well taken. In the new set of experiments, we now include results for a finite dimensional lattice (the rectangular lattice).
>
> Reviewer 4 also points out a significant missing link: we never connected the idea of finite lattices and Gaussian approximate posteriors!! ("...in fact the words "gaussian" or "normal" don't even appear in 2.4..."). We thank the reviewer for finding this gap; we couldn't see it because we are too close to the subject matter. As it turns out, very good lattices not only look like spheres, a uniform distribution over their cell convergences to that of a Gaussian in a KL divergence sense. This is not a new fact - it is well known in the theory of lattices but we never actually mentioned this. The new version of the paper corrects this omission. We also took advantage of this iteration to remove the high resolution analysis to the appendix, since it is not necessary for a VAE experimental section (it was more important for the autoencoder experiments).
>
> Reviewer 4 also points out that "there is little motivation for using lattices over other discrete latent variable models in the first place (eg. straight-through estimator with Gumbel softmax or VQ-VAE)". Agreed. In the new, simpler experimental section we study a discrete latent variable problem (which is connected to VAEs) instead of the problem of straight auto-encoding; the choice of setup (MNIST) and problem (VAE) makes it much easier to compare to other techniques. Thus one storyline we can now claim goes roughly as follows: "Gaussian VAEs are really good, but not discrete, what if we could provably approximate their performance using discrete latent variable models? Here's how you do it with lattices." This is not something that can be claimed, to the best of our knowledge, by techniques such as Gumbel-Softmax based discrete latent variable models. As far as the relation to VQ-VAEs, we now clarify in the text that lattice representations, when used in a VAE setting, can be seen as a form of a VQ-VAE in which the vector quantizers come from a structured codebook instead of being specified as free parameters in a machine learning model.
>
> Finally, reviewer 4 suggests reordering content so  that the problem is introduced first, and then the solution, instead of the other way around. Point taken; the updated version reflects this.
>
> On the smaller errors: we've corrected most of them. The exception is that we couldn't precisely locate the difference in definitions for $G_1$ that reviewer 4 suggests may exist. Thank you!!!

---

### Decision · Program_Chairs · 2019-12-19

**Decision:**

Reject

**Comment:**

This paper presents a new view of latent variable learning as learning lattice representations.

Overall, the reviewers thought the underlying ideas were interesting, but both the description and the experimentation in the paper were not quite sufficient at this time. I'd encourage the authors to continue on this path and take into account the extensive review feedback in improving the paper!